# Effect of Different Preparation Parameters on the Stability and Thermal Conductivity of MWCNT-Based Nanofluid Used for Photovoltaic/Thermal Cooling

Miqdam T. Chaichan [1], Hussein A. Kazem [2,3], Moafaq K. S. Al-Ghezi [4], Ali H. A. Al-Waeli [5], Ali J. Ali [6], Kamaruzzaman Sopian [7], Abdul Amir H. Kadhum [8], Wan Nor Roslam Wan Isahak [9], Mohd S. Takriff [10] and Ahmed A. Al-Amiery [1,9,*]

1    Energy and Renewable Energies Technology Research Center, University of Technology-Iraq, Baghdad 10066, Iraq
2    Faculty of Engineering, Sohar University, P.O. Box 44, Sohar 311, Oman
3    Solar Energy Research Institute, Universiti Kebangsaan Malaysia, Bangi 43600, Selangor, Malaysia
4    Mechanical Engineering Department, University of Technology-Iraq, Baghdad 10066, Iraq
5    Engineering Department, American University of Iraq, Sulaimani 46001, Iraq
6    Department of Biomedical Engineering, University of Technology-Iraq, Baghdad 10066, Iraq
7    Department of Mechanical Engineering, Universiti Teknologi PETRONAS, 32610 Seri Iskandar, Perak Darul Ridzuan, Malaysia
8    Faculty of Medicine, University of Al-Ameed, Karbala 56001, Iraq
9    Department of Chemical and Process Engineering, Faculty of Engineering and Built Environment, Universiti Kebangsaan Malaysia (UKM), Bangi 43000, Selangor, Malaysia
10   Chemical and Water Desalination Engineering Program, Department of Mechanical & Nuclear Engineering, College of Engineering, University of Sharjah, Sharjah 26666, United Arab Emirates
*    Correspondence: dr.ahmed1975@gmail.com or dr.ahmed1975@ukm.edu.my

**Abstract:** The thermal conductivity and stability of any nanofluid are essential thermophysical properties. These properties are affected by many parameters, such as the nanoparticles, the base fluid, the surfactant, and the sonication time used for mixing. In this study, multi-walled carbon nanotubes (MWCNTs) were selected as additive particles, and the remaining variables were tested to reach the most suitable nanofluid that can be used to cool photovoltaic/thermal (PVT) systems operating in the harsh summer conditions of the city of Baghdad. Among the tested base fluids, water was chosen, although ethylene glycol (EG), propylene glycol (PG), and heat transfer oil (HTO) were available. The novelty of the current study contains the optimization of nanofluid preparation time to improve MWCNTs' PVT performance with different surfactants (CTAB, SDS, and SDBS) and base fluids (water, EG, PG, and oil). When 1% MWCNT mass fraction was added, the thermal conductivity (TC) of all tested fluids increased, and the water + nano-MWCNT advanced all TC (EG, PG, and oil) by 119.5%, 308%, and 210%, respectively. The aqueous nanofluids' stability also exceeded the EG, PG, and oil at the mass fraction of 0.5% MWCNTs by 11.6%, 20.3%, and 16.66%, respectively. A nanofluid consisting of 0.5% MWCNTs, water (base fluid), and CTAB (surfactant) was selected with a sonication time of three and quarter hours, considering that these preparation conditions were practically the best. This fluid was circulated in an installed outdoor, weather-exposed PVT system. Experiments were carried out in the harsh weather conditions of Baghdad, Iraq, to test the effectiveness of the PVT system and the nanofluid. The nanofluid-cooled system achieved an electrical efficiency increase of 88.85% and 44% compared to standalone PV and water-cooled PVT systems, respectively. Additionally, its thermal efficiency was about 20% higher than that of a water-cooled PVT system. With the effect of the high temperature of the PV panel (at noon), the electrical efficiency of the systems was decreased, and the least affected was the nanofluid-cooled PVT system. The thermal efficiency of the nanofluid-cooled PVT system was also increased under these conditions. This success confirms that the prepared nanofluid cooling of the PVT system approach can be used in the severe weather of the city of Baghdad.

**Keywords:** SWCNTs; photovoltaic/thermal; stability; thermal conductivity; sonication time; surfactant

## 1. Introduction

Heat transfer from a hot to a colder place is an important, challenging, and needed application in many ways, including for power generation, industry, production processes, chemical industries, vehicles, microelectronics, food industries applications, etc. Improving the heat exchange performance of any application in the sense of reducing the time required for heat transfer will reduce the processing time, increase the life of the equipment, and save energy [1]. An example of a heat exchanger is a radiator in cars. The improvement of heat transfer means the use of smaller heat exchangers to cool the engine water, as a result reducing the weight of the car [2], which means reducing fuel consumption and the consequent reduction in emissions [3].

For many years, traditional fluids such as water, oil, ethylene glycol, etc., have been relied on as heat transfer fluids in industrial and commercial applications. These fluids have performed their duties as best they can. These fluids have a low thermal conductivity, which hinders the rapid transfer and disposal of heat. The negative property of these fluids has caused a limitation in their use in dynamic transfer (fast charging and discharging processes) applications [4]. Micro-sized particles were added to these fluids with high conductivity and this addition improved the thermal conductivity of the product. Most of the added particles were either metals or metal oxides. However, these fluids suffered from low stability, meaning the speed of particles gathering and depositing at the bottom of the container [5], in addition to causing corrosion of components, blockage of narrow passages, and low pressure of the flowing fluid [6]. After the emergence of nanoparticles, the interest shifted to developing nanofluids with high thermal conductivity by adding different types of nanoparticles to the base fluid (usually one of the conventional fluids). These nanofluids attracted the interest of manufacturers and researchers because they enhance thermal conductivity in liquids to which they are added in a small proportion. With this addition, the thermophysical properties of the emulsion are affected, and the result is remarkable [7]. Precisely prepared nanofluids do not cause a decrease in flow pressure and significantly improve heat transfer properties [8]. Therefore, preparing the nanofluid to be stable for a long period is a prerequisite for using this fluid in heat transfer applications. It is also an important requirement in maintaining equipment and raising its efficiency.

The strategic shift of the countries of the world today towards environmentally friendly renewable energies to reduce global environmental risks such as global warming and climate change has caused an increase in the share of renewable energy in the production of electric energy at the expense of fossil fuels (the cause of the problems mentioned). It should be noted that solar energy has begun to occupy this position in most parts of the world [9]. It is well known that photovoltaic cells are among the solar energy applications that have occupied their place among the alternatives for generating renewable electricity. Interest in installing and operating PV stations all over the world is escalating due to the growing interest in environmental cleanliness. This is provided by PV systems available around the world operating on clean fuel, which is the sun's rays.

It is worth mentioning that there are many factors driving the solar energy transition, such as incentives, policy, regulations, behavior, and sustainability [10–12]. Many parameters affect PV panel performance, such as technology, design, irradiance, dust, humidity, ambient temperatures, and other environmental parameters [13,14]. These cells are affected by many environmental influences, such as shadows, temperature, relative humidity, and dust [15–18]. Theoretically, the operation of PV modules under standard conditions (solar radiation intensity of 1000 W/m$^2$, air temperature of 25 °C, and air mass of 1.5) results in the highest electrical efficiency. So, the higher the solar radiation, the more electricity the PV module generates. Practically speaking, it is not at all like this. As the bulk of the solar radiation is absorbed by the cell to increase its temperature, the smaller part goes to generate electricity. High cell temperature causes a decrease in the generated power and



a deterioration in the electrical efficiency of the system [13]. The researchers proposed to reduce the negative effects of this thorny issue (since the best fields for creating photovoltaic fields are in deserts with high solar radiation) by using PVT systems [19].

PVT systems are solar collectors consisting of a photovoltaic panel connected to a thermal solar storage tank aimed at withdrawing the heat collected in the solar panel and cooling it to improve its electrical efficiency and benefit from this withdrawn heat in thermal applications [20]. The increase in the temperature of the solar panel causes the deterioration of the generated electrical power, while its cooling causes the improvement of this productivity. PVT systems can be cooled with water or different nanofluids, many of which have been tested [21].

The thermal conductivity and stability of the nanofluid are the two main characteristics affecting heat transfer efficiency. Studies have shown that SWCNTs and MWCNTs have very high thermal conductivities that are not comparable to any metallic or metal oxide nanoparticles known to date. Usually, most researchers are not interested in the details of preparing the nanofluid as they adopt one procedure for all the prepared nanofluids. Here is the question: What proves that the fluids have taken their right of preparation? Meaning, each fluid (depending on the type of nanoparticles and the added mass fraction) needs preparation time and treatment care that differs from other fluids. In most of the studies presented, no attention is shown for this issue.

Nanofluid preparation is not sufficient to reach the optimal performance of a nanofluid-cooled PVT system. There are many parameters, such as the heat exchanger design and weather conditions (such as solar radiation intensity and ambient temperature). In this study, a selected nanofluid was prepared from several experiments to reach the best base fluid and added mass fraction, the best surfactant, and the most suitable preparation time using ultrasonic vibration. This fluid will be experimentally tested in the PVT system to compare the resulting performance with a water-cooled system to show whether the added costs of the PVT system can be recovered through the gains in electrical and thermal performance. The experiments were carried out in the most severe weather conditions in the city of Baghdad during the hottest months of July and August. The success of any system in the conditions in which the experiments were carried out means its success for the rest of the year.

## 2. Literature Review

The addition of nanoparticles to the base fluid causes obvious changes in the physical properties of the produced emulsion, such as fluid color, density, and viscosity, in addition to its thermal properties such as its thermal conductivity and heat capacity [22]. These properties depend directly on the properties of the added nanoparticles, and they are numerous, including the crystal structure of the molecules, surface-to-volume ratio, surface curvature, diffusivity, catalytic activity [23], electrical resistance, etc. [24]. The properties of the base fluid also have a role in determining the thermophysical properties of the emulsion [25]. Many research studies conducted in the literature involve researchers using many types of nanoparticles to create nanofluids for various engineering uses. To date, there has been no agreement on a specific type of these additives or the base fluid, meaning there is no uniformity around an ideal nanofluid for use in an application. The most important properties that determine whether a nanofluid is close to ideal or far away are two important properties: the thermal conductivity of the nano-suspension and its stability [26].

Since the discovery of nanoparticles, researchers have relied on preparing nanofluids from nanoparticles of metallic origins such as gold, silver, and copper, which have high thermal conductivities [27]. Then, they found that the use of nanoparticles of metal oxide origin is cheaper and has conductivity comparable to the first set. Carbon nanotubes have also appeared as additives for forming nanofluids with excellent thermophysical properties [28]. Carbon nanotubes presented distinct and unique thermal properties, and the nanofluids formed from them were considered to have superior thermal capabilities [29].

Carbon nanotubes are cylindrical particles with diameters ranging from one nanometer to several nanometers and a cylinder length of several micrometers. These tubes are graphene sheets rolled into a cylindrical shape. These tubes are divided into single-walled carbon nanotubes (SWCNTs) and multi-walled carbon nanotubes (MWCNTs) depending on the treatment [30]. Both types have unusual properties in heat transfer, as they have a high thermal conductivity (2000–6000 W/m$^2$ K) that exceeds hundreds of times the nanoparticles found [31], whether metallic or metal oxides [32]. Carbon nanotubes in conventional heat transfer fluids completely disperse and raise their thermal conductivity [33] compared to the base fluid [34]. Research studies on nanofluids, in general, and on CNTs' nanofluids, in particular, have developed and studied various fields related to them in order to improve their effectiveness in potential applications, whether industrial or civil. Carbon nanotube fluids can improve bubble adsorption in the heat-driven absorption system [35] and enhance heat transfer in heat exchangers and solar thermal collectors [36]. This is in addition to reducing the effect of Leidenfrost in the cooling process [37].

For any nanofluid, its thermal conductivity depends on the properties of the base fluid and the nanoparticles added to it. The characteristics of the nanoparticles dispersed in the base liquid include their crystal structure and the shape and size of the molecule. Additionally, the factors for forming the nanofluid include the additive mass or volume fractions, the surfactant concentrations [38], and the interactions that occur between the added nanomaterials and the basic liquid, etc. [39]. These factors affect the thermal conductivity and stability of carbon nanofluids in different but significant proportions. SWCNTs and MWCNTs have been extensively studied and used to form nano-emulsions with many different base liquids. These two types have been used in many heat transfer applications, and interest in them increased with the dawn of the cooling of solar cells or so-called photovoltaic–thermal (PVT) systems.

Xing and Wang (2015) compared the effect of adding three types of carbon nanotubes to water on the thermal conductivity of the experimentally produced liquids [40]. The thermal conductivity of the prepared emulsions improved compared to the base liquid, and this conductivity increased with the increase in the concentrations of the CNT particles. The results of the study showed that the addition of SWCNT particles with short and long cylinder lengths and MWCNT particles at a concentration of 0.48% (by volume) enhanced the thermal conductivity of the prepared emulsions by 8.1%, 16.2%, and 5.0%, respectively, at a fluid temperature of 60 °C. The researchers concluded that preparing a nanofluid by adding long SWCNT particles to water gives the highest thermal conductivity. Additionally, the relationship between the improvement in the thermal conductivity of the produced nanofluid and the increase in the concentration of carbon nanotubes and the operating temperature under the tested conditions is almost linear. Vankatesh et al. (2022) prepared several water based graphene nanofluids with different concentrations of nanoparticles to test their effect on the performance of PVT systems experimentally [41]. The concentrations selected by the authors were 0, 0.1, 0.2, and 0.3 (by volume %). The performance of the PVT system improved by using the prepared nano-emulsions as the efficiency of the systems increased, compared to cooling them with water. The results showed that graphene nanoparticles showed high cooling effects for PVT systems as the panel temperatures were reduced by 20 °C compared to water cooling PVT system.

A cooling PVT system was studied using nanofluids prepared by adding SWCNTs (with four weight ratios of 0.1%, 0.5%, 1.0%, and 2.0%) to a base fluid (which was a mixture of water with a volume of 75.0% and ethylene glycol of 25%) as claimed by Kazem et al. in 2021 [42]. The researchers selected the nanofluid prepared by adding 0.5% SWCNTs to cool the PVT system after several experiments to evaluate its thermophysical properties. The addition ratio was chosen because the prepared nano-emulsion improved the thermal conductivity by 103% and had excellent stability that exceeded 109 days when tested by the camera images and, according to the zeta potential, reached 65 mV. The proposed emulsion caused a significant increase of 11.7% in the generated electric power and 25.2% in electrical efficiency compared to a standalone PV system.

Nanofluids' stability has an essential role in maintaining a safe and stable heat transfer process in the application in which it is used. The stability of the nanofluid means that the nanoparticles within it remain dispersed, distributed, and not agglomerated. The researchers used various techniques and methods to improve the dispersion of nanoparticles in the base liquid. As surfactants were used, ultrasonic vibration technology was also adopted. These two techniques have become widespread and have been used in many research studies [43]. Duangthongsuk and Wongwises (2010) placed the mixture of water and $TiO_2$ nanoparticles in an ultrasonic vibrator and treated the nanocluster with ultrasound for two hours [44]. As for Wang et al. (2009), this technique was used for a mixture of nano-$Al_2O_3$ and water for 15 min [45]. Asadi (2020) used the same previous process for one hour to mix nano-MWCNT and water and to reach high stability [46]. Ultrasound treatment that breaks up the bonds between the nanoparticles and scatters and distributes them homogeneously throughout the container improves the stability property of the nanofluids.

Researchers have not yet agreed on the optimal time to use sonication to disperse nanoparticles in the base fluid. For example, Lee et al. (2014) used an ultrasonic treatment of $Al_2O_3$–water fluid for more than 5 h and concluded that the effect of long sonication time is negative on the thermal conductivity and stability of the nanofluid [47]. Mahbubul et al. (2015) studied the effect of sonication on the stability of the nanofluid and concluded that increasing the sonication time for more than one hour did not show an improvement in the stability of the nanofluid [48]. Dhahad and Chaichan (2020) adding 50 and 100 ppm of nano-$Al_2O_3$ and nano-ZnO to diesel and mixed them in an ultrasonic container [49]. The results showed the stability of the fluids produced for 76 and 81 days for nano-ZnO and nano-$Al_2O_3$, respectively, when added at a concentration of 50 ppm. When the concentration was increased to 100 ppm, the stability of the emulsions decreased to 68 and 72 days for nano-ZnO and nano-$Al_2O_3$, respectively. Habib et al. (2021) added SWCNT particles of 0.1%, 0.5%, 1%, 3%, and 5% (by weight%) to molten paraffin wax (at 80 °C to avoid wax hardening) with ultrasonic shaking for 2 h [50]. The researchers also kept the prepared suspensions by sonication at a temperature of 65 °C for 24 h in a special oven to maintain the stability of the SWCNTs and paraffin mixture.

Therefore, in this study, many nanofluids prepared by adding multiple mass fractions of MWCNTs to the base fluid are tested. Additionally, in this study, many base fluids such as water, ethylene glycol (EG), propylene glycol (PG), and oil are tested. The addition of various surfactants to the above-mentioned base fluids are also tested. The novelty of the current study contains the optimization of nanofluid preparation time to improve MWCNTs' PVT performance with different surfactants (CTAB, SDS, and SDBS) and base fluids (water, EG, PG, and oil). In the third part of the study, the best sonication time to be used with the constituent of water and the above additives are tested. Finally, the best ratio of added MWCNTs to the best base fluid (which gives the highest stability and thermal conductivity) for use in PVT systems is determined. The selected final nanofluid is tested practically in cooling a PVT system that operates in harsh weather conditions. The aim of this study is to provide special attention to the methods of preparing nanofluids and the method of selecting the optimal fluid for work based on the enhancement rates it introduces in both thermal conductivity and stability properties.

## 3. Materials and Methods

### 3.1. Base Fluids

In this study, several base fluids were used, which are water, ethylene glycol, propylene glycol, and oil, and the specifications of which are listed in Table 1. The four liquids do not have similar thermophysical properties and were chosen because they are available and affordable. They have been used for many years as heat transfer liquids. These fluids are characterized by low thermal conductivity. The data listed in Table 1 show that all the studied conventional liquids have low thermal conductivity (TC). The table manifests that water has the highest TC among them. However, for the viscosity, water is the least viscous

of these liquids, and the highest is ethylene glycol. In terms of specific heat, which is an important property in heat transfer processes, water is superior to all other liquids in this, followed by EG. The table shows that the surface tension of water is the highest among the listed liquids, and this characteristic is important when adding a surfactant. All the listed specifications were measured in the Chemical Engineering Department, University of Technology-Iraq.

**Table 1.** Base fluid specifications.

| Specifications | Water | Ethylene Glycol | Propylene Glycol | Heat Transfer Oil (HTO) |
|---|---|---|---|---|
| Viscosity (mPa.s) at 25 °C | 1.002 | 1.161 | 0.09 | 1.5 |
| Density (kg/m$^3$) | 1000 | 998 | 1036 | 855 |
| Thermal conductivity (W/m K) | 0.57 | 0.258 | 0.147 | 0.134 |
| Specific heat (J/(kg K) | 4.2 | 2.433 | 0.895 | 2.097 |
| Surface tension (mN/m) | 76.5 | 48.6 | 45.6 | 35 |

*3.2. Surfactants*

A surfactant is used to reduce the surface tension of the fluid in which it is dissolved, thus facilitating its absorption and dissolution in this fluid. This term (surfactant) means a surface-active agent. Surfactants are amphiphilic molecules that can be absorbed in the vicinity of air and water. In this region (the surface of the liquid), the hydrophobic part of these molecules lines up on the air side, and the other (hydrophilic) part is lined up on the water side. As a result, a decrease in surface tensions occurs. Three types of surfactants (Table 2) were also used in the experiments. These surfactants were used in research conducted by Mohd Saidi et al. in 2022 [51]. These surfactants were selected for their availability in local markets at a reasonable cost and for being one of the most commonly used types in research works that dealt with nanofluids. The surfactants' specifications, listed in Table 2, were supplied by the manufacturers.

**Table 2.** Surfactant specifications.

| Specifications | SUR. I | SUR. II | SUR. III |
|---|---|---|---|
| | Cetyltrimethylammonium Bromide (CTAB) | Sodium Dodecyl Sulfate (SDS) | Dodecylbenzenesulfonate (SDBS) |
| Manufacturer | Fisher Scientific UK | Fisher Scientific UK | Fisher Scientific UK |
| Chemical structure | $C_{19}H_{42}BrN$ | $C_{12}H_{25}NaO_4S$ | $C_{18}H_{29}NaO_3S$ |
| Electrical conductivity ($\mu s\ cm^{-1}$) | 94.9 | 65 | 68 |
| Turbidity (NTU) | 0.095 | 0.045 | 0.030 |
| pH | 6.13 | 9.1 | 8.5 |
| Molecular weight (g/mole) | 464.45 | 288.5 | 348.48 |
| Density (g/cm$^3$) | 0.5 | 1.01 | 0.18 |

The ease of nanoparticle agglomeration in nano-suspension is due to the high surface energy of these particles [4]. The agglomerated and then deposited particles will lead to a decrease in the thermal conductivity of the nanofluid and thus deteriorate the heat transfer process. This process is called a decrease in the stability of the nanofluid, so it is important to investigate it carefully to determine the stability of the nanofluid. The addition of surfactants is meant to form weak electrostatic interactions with the target nanoparticles [52] and to stabilize the non-covalent interaction in the suspension [53]. The nanofluid prepared by adding an amount of surfactant is assumed to have higher stability [54]. Here, the concentration of the surfactant affects the stability of the product increase or decrease rates. Studies have proven that the highest stability that can be reached is at pH 4 [55], and the lowest stability is at pH 10 [56]. Here, the type of surfactant, whether anionic or non-ionic, plays an important role. Ionic surfactants degrade the stability of

the nanofluid, and conversely, cationic and non-ionic surfactants improve stability [30]. Therefore, the addition of an alkaline surfactant causes a significant decrease in the stability of the nano-suspension. As Table 2 shows, the pH of SUR I is more acidic than the other two types, so its effect on the stability of the formed nanofluids will be the best. The nanofluids reusability and stability are critical for sustained operation. The purpose of the proposed system is to operate daily and to circulate with minimal losses. Moreover, the cost of producing such a nanofluid and its integration into the system makes it crucial to minimize evaporation and leakage losses. The type of container and pipes must therefore be selected with consideration for their permeability. Another element that is considered is the impact of operational conditions on the long-term stability of the nanofluids. This issue represents one factor out of many other operational factors that will impact the overall performance of the system. It is important to note that the standardization of methods, quantities, and mixing requirements should help in producing nanofluids with similar thermophysical properties.

### 3.3. MWCNTs

As for the MWCNTs with long cylinders, the details are listed in Table 3. An MWCNT can be represented as a long, coiled graphene sheet with a length-to-diameter ratio of 1000. The internal diameter of these tubes generally does not exceed 5–15 nm, while the external one ranges 8–30 nm. The used nano-tubes' length is $\geq$20 μm, so they can be considered one-dimensional structures. These tubes have distinct properties compared to metal nanoparticles or metal oxides. MWCNTs are excellent conductors with a thermal conductivity of about 2000 to 4000 W/m K [57]. These tubes are blamed for the fact that the cost of their production is still high, and work is still underway to develop production techniques at affordable costs. The MWCNTs' specifications, listed in Table 3, were supplied by the manufacturers. The price of 1 gram of the used MWCNT in Iraq markets is about USD 5. This price is higher than many nano metal oxides costs. However, the gained TC is much higher than what the metal oxide nanofluid introduce. In such a case, the cost will be offset by using lower mass fractions of MWCNTs, which reduces the prepared nanofluid cost to be comparable to the metal oxide fluids.

**Table 3.** MWCNT specifications.

| Manufacturer | Carbon Nanomaterial Technology (South Korea) |
| --- | :---: |
| Diameter (external) (nm) | 8–30 |
| Diameter (internal) (nm) | 5–15 |
| Tube length (μm) | $\geq$20 |
| Number of walls | 3–10 |
| Assay | $\geq$95 wt.% |
| Form | Powder |
| Amount of impurities (wt.%) | $\geq$5 |
| Bulk density (g/cm$^3$) | 0.25–0.35 |
| Surface area (m$^2$/g) | $\geq$270 |
| Melting point (°C) | 3670 |
| Thermal conductivity (W/m K) | 3000 |
| Thermal stability in air (°C) | $\geq$600 |

### 3.4. Nano-Emulsion Preparation

The ultrasonic vibration technique was used to prepare all the nano-emulsions. In the first part of the experiments, the base fluid was mixed with the nanoparticles in previously determined mass fractions of 0.1, 0.5, 0.75, and 1.0. The prepared emulsions were subject to thermal conductivity and stability (ST) examinations, and the rest of the thermophysical properties (density and viscosity) have not been examined because the changes that occur in them can be neglected, as indicated by [58]. In the second set, 0.5% (wt.%) of the tested surfactants was added to the prepared emulsions. The TC and ST of the prepared emulsions were tested, and the best nano-emulsion was selected. In the third set, this

selected nano-emulsion was prepared again using the selected base fluid and surfactant but with variable sonication time. Samples were tested at six different times (1.5, 2, 2.5, 3, 3.15, 3.5, and 4 h). The prepared emulsions in this set were also subjected to TC and ST tests to evaluate the best sonication time for the tested materials. After choosing the appropriate base fluid, surfactant, sonication time, and added MWCNT mass fraction, this nanofluid is used in cooling a PVT system to evaluate its activity and to compare it with other works in the literature.

### 3.5. Instrumentations

In this study, to achieve the best and most accurate results, an ultrasonic vibrating mixer (TELSONIC ULTRASONICS CT-I2) was used. The added weights were verified using an accurate digital METTLER TOLEDO scale (USA-made) that measures up to 1/10,000th of a gram. The thermal conductivity of the prepared nanofluid was measured with a KD2 Pro analyzer scale (ICT International, India). As for the stability of the prepared fluids, this was measured using a Nano Zetasizer (ZSN) (GmbH). Each set of experiments and measurements was repeated three times as a way to confirm the repeatability of experiments and reduce measurement uncertainty. Each instrument was calibrated before its use and its accuracy was determined. These values were used to determine the uncertainty, the details of which are listed in Table 4. The following equation shows the total uncertainty of the experiments [40]:

$$e_R = \left[ \left( \frac{\partial R}{\partial V_1} e_1 \right)^2 + \left( \frac{\partial R}{\partial V_2} e_2 \right)^2 + \cdots + \left( \frac{\partial R}{\partial V_n} e_n \right)^2 \right]^{0.5} \tag{1}$$

where $e_R$, $R$, $e_i$, and $\frac{\partial R}{\partial V_1}$ represent the results uncertainty, independent variable's function, the uncertainty interval in the $n_{th}$ variable, and single variable measured result sensitivity, respectively. Table 4 lists the instruments used and their uncertainties. The total test instrumentation uncertainty was 1.933, which reveals acceptable an accuracy of the measuring devices. A DC electronic load 3711A device was used to measure short-circuit current, open-circuit voltage, and maximum power. The device measurements were verified by comparing to the meter's measurements.

$$e_r = \left[ (0.78)^2 + (0.47)^2 + (0.55)^2 + (0.032)^2 + (0.79)^2 + (1.26)^2 + (0.54)^2 \right]^{0.5} = \pm 1.933 \tag{2}$$

**Table 4.** Uncertainties of the used instruments.

| Equipment | Parameter | Experimental Uncertainty |
|---|---|---|
| KD2 Pro analyzer | Thermal conductivity | ±0.78% |
| Nano Zetasizer | Zeta potential | ±0.47 |
| Multi-meter | Voltage | ±0.55% |
| Multi-meter | Current | ±0.032% |
| Luminous intensity meter | Irradiance | ±0.79% |
| Thermocouples | Temperature (PV module, PVT collector, inlet, outlet, and ambient) | ±1.26 % |
| Flow meter | Coolants' flow rate (kg/s) | ±0.54% |

### 3.6. PVT System Description

After completing the experiments to select the most suitable nanofluid for use in PVT systems, this fluid was circulated in a system prepared for this purpose. The direct flow absorber was chosen for the nanofluid circulation due to its ease of manufacture and low cost compared to other types of collectors. Figure 1 shows a schematic diagram of the system used. The PVT system consists of a PV module mounted on the back by a single-channel direct-flow absorbent to circulate the selected nano-emulsion. Two PVT systems were used, one cooled by water and the second cooled by the prepared nanofluid.

Three monocrystalline type PV modules were used, and their specifications are listed in Table 5. Each module's width is 0.65 m and its length is 1 m.

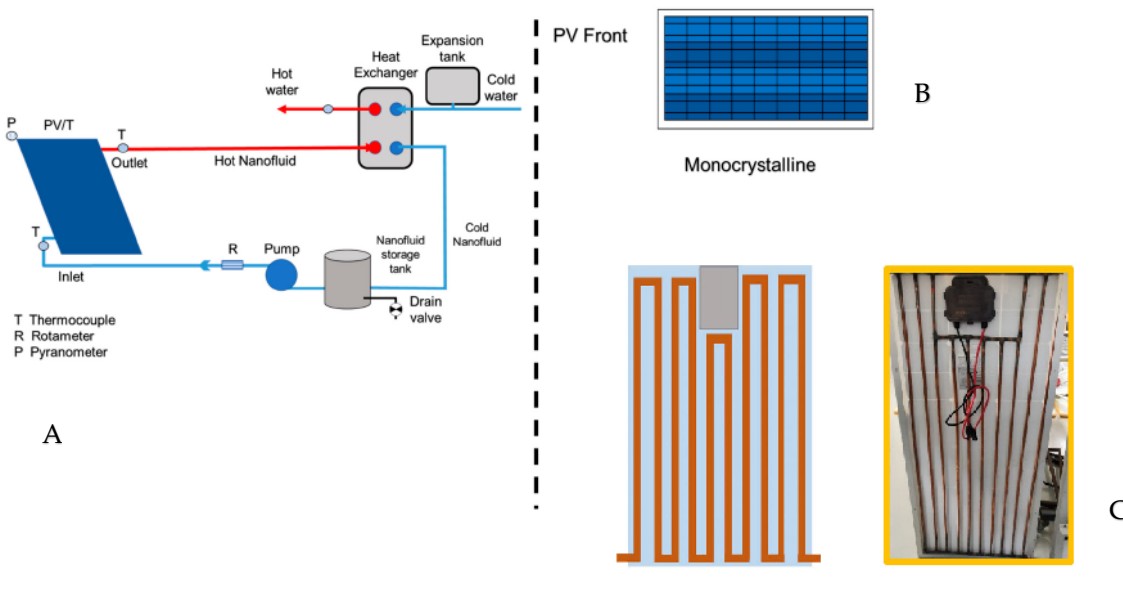

**Figure 1.** Schematic drawing of the PVT system used in tests with a direct flow heat exchanger. (**A**) System drawing, (**B**) PV front, and (**C**) direct flow heat exchanger drawing and picture.

**Table 5.** Module specifications.

| Solar Module Type | Nutu Tech Fzco |
| --- | --- |
| Peak power | 100 W |
| Max. power voltage | 17.96 V |
| Max. power current | 5.57 A |
| Open-circuit voltage | 22.6 V |
| Short-circuit current | 5.76 A |
| Weight | 11.4 kg |
| Dimensions | $1010 \times 660 \times 34$ |
| Operating temperature | $-40\,^{\circ}$C to $90\,^{\circ}$C |
| Wind resistance | 2400 Pa |

The practical tests were carried out in the outdoor conditions in the city of Baghdad. Baghdad (the capital of Iraq) suffers from severe weather conditions. The intended weather conditions are continental, very hot in summer (exceeding 60 °C under the sun), and mild in winter (not less than 14 °C during the day). As for the rain, the city has suffered from a decrease in its rainfall during the past three decades, with the rise of dust and frequent dust storms [36]. The used PV panels were oriented at an angle of 33° to the south to give the greatest possible productivity during the day [37]. Table 6 shows the average weather conditions for Baghdad through the study days of July and August 2021, which are considered the hottest months of the year and with increased dust. The voltage and the resulting current from the three studied modules were recorded, seeking from early morning until sunset. To limit the number of tests, a mass flow rate of 0.015 kg/sec was chosen, according to the results of [59]. A tank for circulated nanofluid was used and a TOPSFLO-China pump was employed for the circulation process. The nano-emulsion system is a closed one with valves to control the movement of the coolant.

**Table 6.** Average weather conditions for July and August 2021 for the city of Baghdad.

| Parameters | July | August |
|---|---|---|
| Max. temp (°C) | 51 | 47 |
| Min. temp. (°C) | 35 | 34 |
| Shining hours (hr./day) | 14.5 | 13 |
| Precipitation (mm) | 0 | 0 |
| Rainy days | 0 | 0 |
| Humidity (%) | 44 | 31 |
| Wind speed (m/s) | 3 | 2.5 |

The electrical efficiency of the studied systems and the thermal efficiency of the two PVT systems were calculated using the following equations:

$$\text{The electrical power is}: \ P_{max} = I_{mp} \times V_{mp} \tag{3}$$

$$\text{The useful collected heat (W) is}: \ Q_u = \dot{m} C_p (T_o - T_i) \tag{4}$$

$$\text{The electrical efficiency } (\eta_e) \text{ is}: \ \eta_e = \frac{P_{max}}{I_s \times A_{panel}} \tag{5}$$

$$\text{The system's thermal efficiency is}: \ \eta_{th} = \frac{Q_u}{I_s \times A_c} \tag{6}$$

$$\text{The total efficiency is}: \ (\eta_t) = \eta_t = \eta_{th} + \eta_e = \frac{Q_u + P_{max}}{I_s \times A_t} \tag{7}$$

## 4. Results and Analysis

Several tests were performed to reach the best nanofluid for use in cooling PVT systems. It must be emphasized here that more tests are required to reach such a nanofluid, but the current study reduces the possibilities and tests that must be conducted in the future to reach this goal.

### 4.1. Base Fluid Type Effect

In the first part of the experiments, the base liquid was mixed with the nanoparticles in predetermined proportions of 0.1, 0.5, 0.75, and 1.0 wt.%. The TC of the prepared nanofluids is shown in Figure 2. The TC of water is higher than the rest, as shown in Table 1. All thermal conductivities were improved when MWCNT was added to all tested base fluids. However, the TC of the aqueous origin nanofluid was superior to the rest of the species and for all added ratios. At the same time, TC was improved by increasing the amount of MWCNTs added for all base fluids. When 1% of MWCNTs was added, the improvement values were 21.9%, 18.94%, 23.8%, and 33.33% for water, EG, PG, and HTO, respectively, compared to base fluid TC. At this mass fraction, water TC exceeded EG, PG, and HTO by 119.5%, 308%, and 210%, respectively. These results indicate that using water as a base fluid is the best among the tested fluids.

Figure 3 shows the effect of the base fluid type and the mass fraction of the added SWCNTs on the ST of the prepared nanofluids. In general, all the prepared fluids had a zeta potential higher than 40 mV, which means a high ST. When adding 0.1% MWCNT, the zeta potential exceeded 70 mV for all base fluids tested, which is excellent stability. At all tested added mass fractions, aqueous nanofluids had the highest stability even at high addition rates (1%). When MWCNTs were added by a mass fraction of 0.5%, the water-based nanofluid stability exceeded EG, PG, and oil stabilities by 11.6%, 20.3%, and 16.66%, respectively. However, all these nanofluids are characterized by high zeta potential (more than 60 mV).

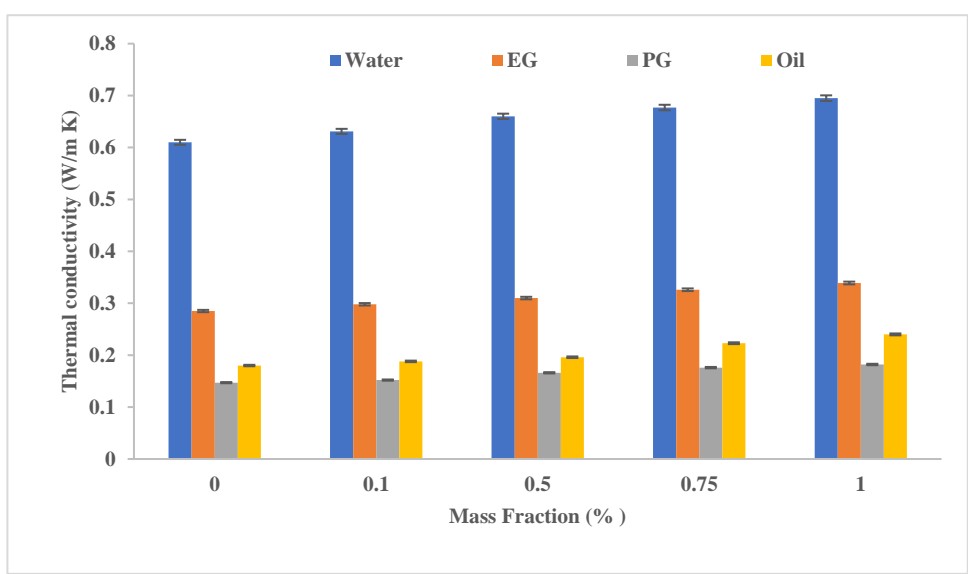

**Figure 2.** The effect of base fluid type and MWCNT (added in variable mass fractions) on the TC of the resulting nanofluid.

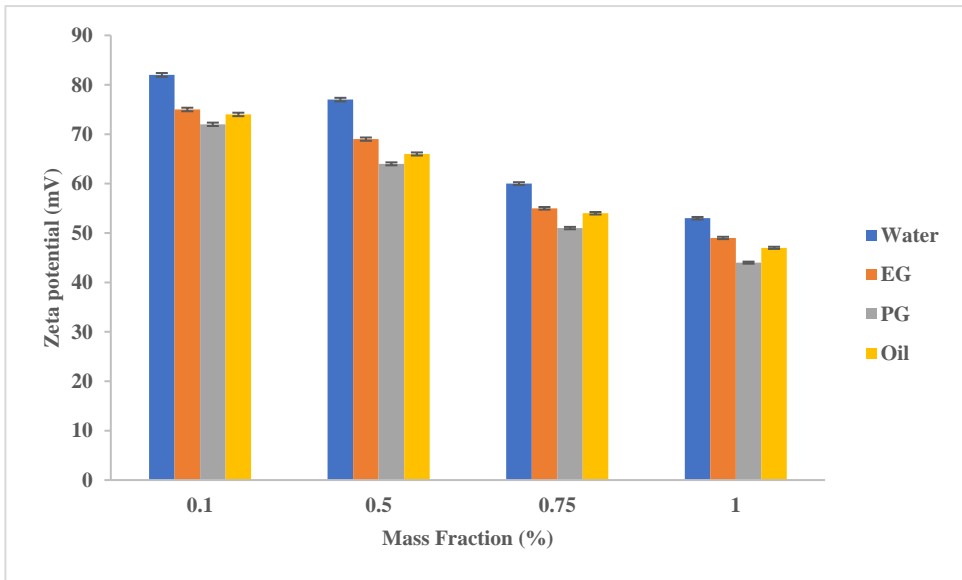

**Figure 3.** The effect of base fluid type and MWCNT (added in variable mass fractions) on the stability (zeta potential) of the resulting nanofluid.

### 4.2. Surfactant Type Effect

In the second set, 0.5% (mass fraction) of the tested surfactants (Table 2) was added to the prepared emulsions, adopted from the results of Mohd Saidi et al. in 2022 [51]. The effect of the added surfactant was very limited on the resulting thermal conductivity, as it did not increase in a noticeable way, as Figure 4 indicated. SUR I showed a clear effect when it was added to the oil, while for the rest of the fluids, its effect was not clear. The increase in the TC of an aqueous nanofluid was 0.64%, 1.5%, 1.6%, and 0.14% for the added mass fractions of 0.1, 0.5, 0.75, and 1.0%, respectively.

Despite the limited effect of the tested surfactants on the resulting TC, its effect on the ST of the resulting fluids was clear, as Figure 5 manifests. The measured zeta potential values of all tested nanofluids increased in varying proportions depending on the type of base fluid and the surfactant added. For example, for water at an MWCNT mass fraction of 0.5%, an enhancement rate of 10.3%, 5.2%, and 6.5% for the addition of SUR I, SUR II, and SUR III was measured, respectively. The results of the tests show that CTAB increased

the stability of any nanofluid added to it more than the rest of the tested surfactants. For example, when 1% of MWCNT was added, the percentage of increase in zeta potential (ZP) was 7.4% and 5.4% for water nanofluids compared to SUR II and SUR III addition, respectively. When SUR I was added to nano-EG, the resulting increments in ZP were 5.7% and 3.77% compared to SUR II and SUR III addition, respectively. When PG was used, the resulting increments in ZP were 2% and 4.15% compared to SUR II and SUR III addition, respectively. Lastly, when HTO was used, the resulting increments in ZP were 10.2% and −1.5% compared to SUR II and SUR III addition, respectively. One of the main reasons for SUR I's superiority may be its acidic nature, as shown in Table 2. The results of the previous two paragraphs show that using water as a base fluid and CTAB as a surfactant gives the highest TC and ST.

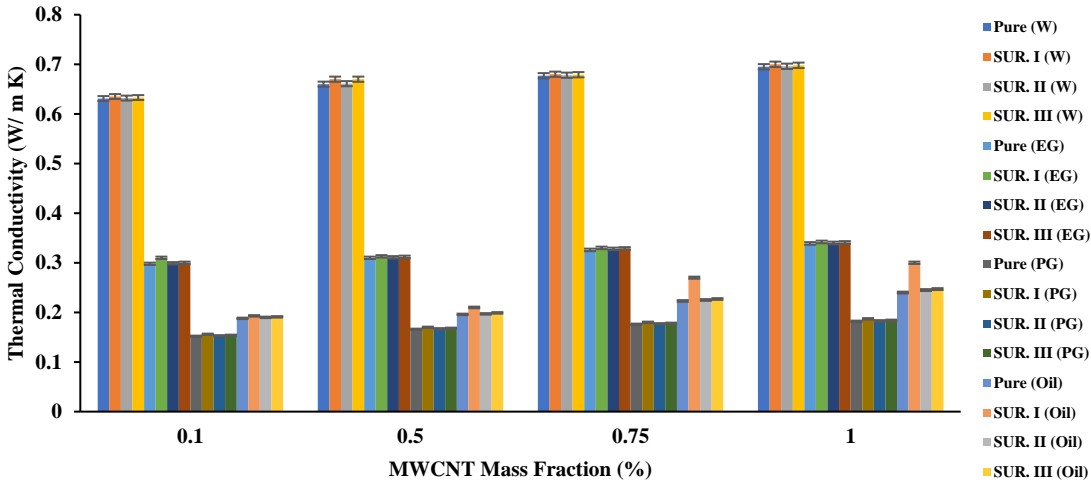

**Figure 4.** The effect of base fluid and surfactant type and MWCNT (added in variable mass fractions) on the TC of the resulting nanofluid.

### 4.3. Sonication Time Effect

In this set of experiments, nano-emulsion consisted of 0.5 (wt.%) MWCNT added to water while employing CTAB as a surfactant. All nanofluids were prepared again using variable sonication time. Seven different timings were tested (1:30, 2:00, 2:30, 3:00, 3:15, 3:30, and 4:00). The prepared emulsions in this set were also subjected to TC and ST tests to evaluate the best sonication time for the tested emulsions. Figure 6 illustrates the effect of sonication time on the TC of the tested nano-emulsions. When MWCNT was added with a mass fraction of 0.1%, the TC increased after two and a half hours of sonication (0.635 W/m K). When the sonication time was increased to three and quarter hours, the TC was decreased to (0.634 W/m K). As for adding 0.5% of MWCNT, the highest TC was at a sonication time of three and a quarter hours and three and a half hours (0.67 W/m K). The TC decreased slightly (0.65 W/m K) when the sonication time was extended to four hours. The best TC for the case of adding 0.75% of MWCNT was at a sonication time of three and a half hours. As for the case of adding 1%, the highest TC was measured at four hours. From the above, it can be concluded that each mixing sample has an optimal sonication time, so this time should be tested with small samples before the quantitative production of the nanofluid. When a small mass fraction is added, it requires less sonication time than adding a large mass fraction of nanoparticles. This is a normal condition, as particles with a small mass fraction disintegrate and spread faster than those with a large mass fraction, which are attracted due to the proximity of the distances between them. Increasing the sonication time, then, the appropriate time causes a loss in TC and also causes losses in costs as the effect of the process is reversed.

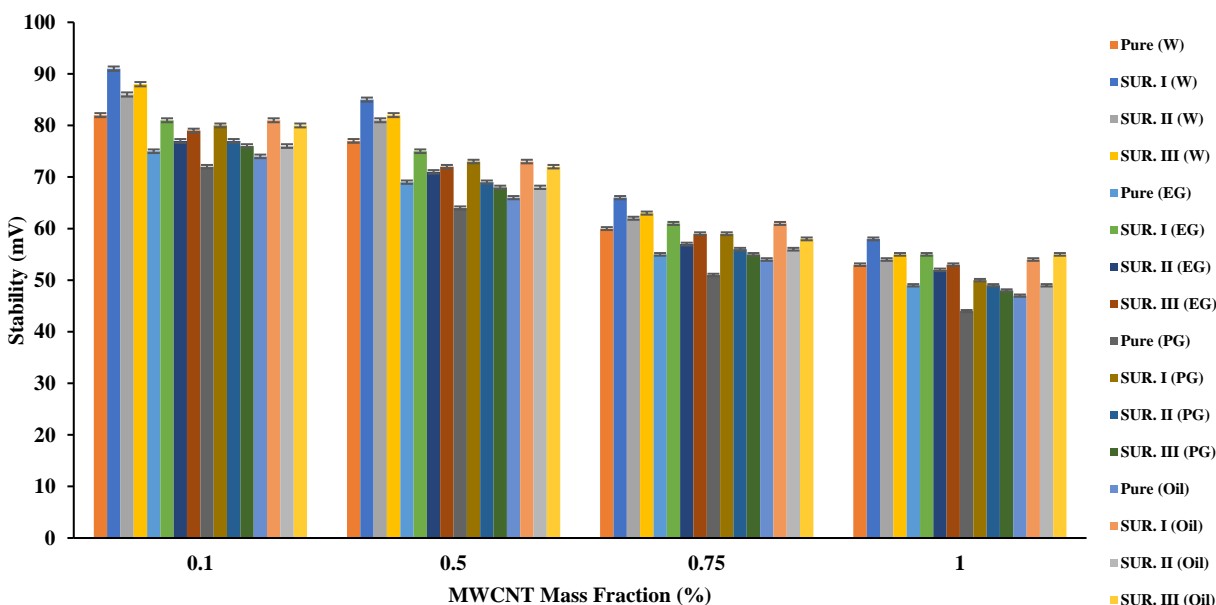

**Figure 5.** The effect of base fluid and surfactant type and MWCNT (added in variable mass fractions) on the stability of the resulted nanofluid.

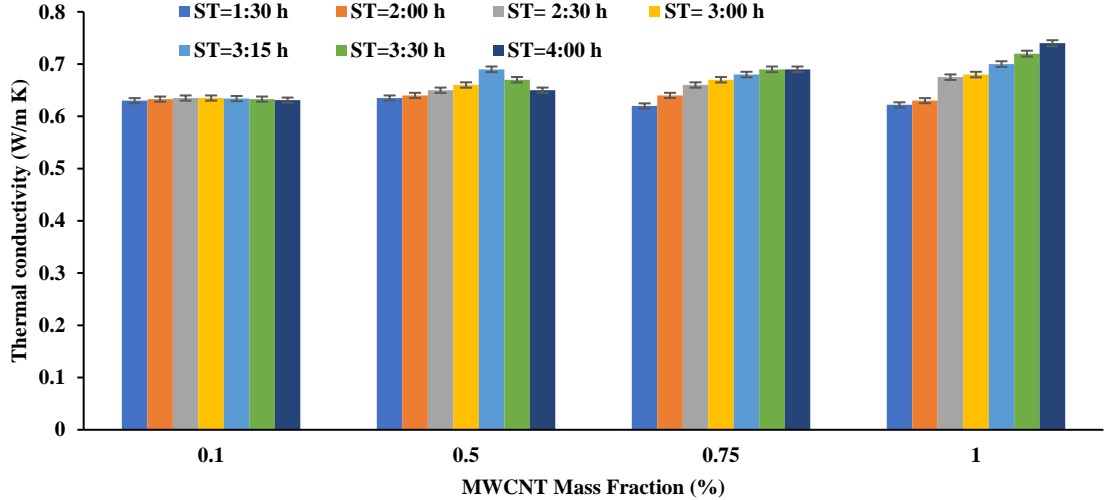

**Figure 6.** The effect of sonication time on the TC of the prepared nanofluids (MWCNT + Water + CTAB).

Figure 7 shows the effect of sonication time variation on the prepared nanofluids' ST. The ST increased with increasing sonication time, reaching its maximum at three and quarter hours for the addition of 0.1% and 0.5%. As for the cases of 0.75% and 1%, the best ST was achieved at a time of four hours. It is noteworthy that there is a relationship between ST and TC, where the ST decreased after reaching the appropriate sonication time, which is almost the same as the result achieved for TC.

From the above results, water can be considered the preferred base fluid that provides the best TC and ST. Additionally, CTAB caused the best results in terms of TC and ST, so it is preferred. As for which MWCNT mass fraction should be added, 0.5% was selected to reduce the cost of nanoparticles and the sonication process. Additionally, the ST provided by this mass fraction nanofluid is excellent, and the TC that differs from the 1% mass fraction is relatively low. So, in the next set of tests, the used nanofluid consisted of 0.5% MWCNT, water, and 0.5 wt.% CTAB.

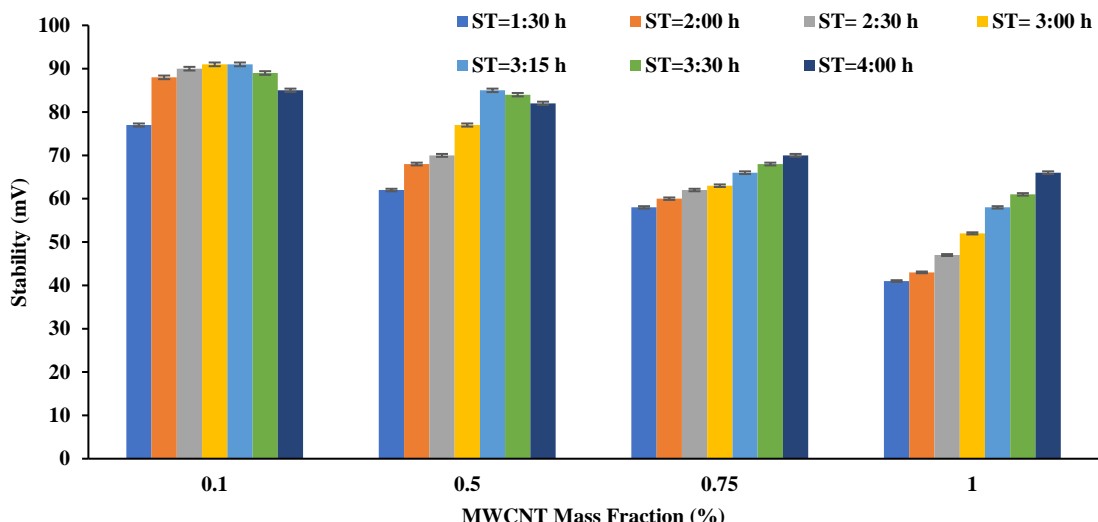

**Figure 7.** The effect of sonication time on the ST of the prepared nanofluids (MWCNT + Water + CTAB).

### 4.4. Outdoor Tests Environmental Conditions

The current study was carried out in Baghdad, Iraq, and it was chosen to work on cooling a PVT system with the prepared nanofluid in the most severe weather conditions. Therefore, the months of July and August were chosen for the examinations. These two months are characterized by the highest yearly temperatures, as the average temperature in the shade exceeded 50 °C most of the study days at noon. Moreover, the average solar radiation intensity exceeded 1000 W/m² at noon. Figure 8 shows the measurements of solar radiation, atmospheric temperatures, and surface temperatures of the PV panel for a standalone module and the two PVT systems cooled by water and nanofluid. The temperature of the PV panels was raised to a maximum of 78 °C at noon (2:15 pm). At this time, the ambient temperature is only 50 °C. Here, it must be emphasized that the ambient temperature is measured in the shade while the temperature of the PV panel surface measured is completely exposed to the sun. The results of the figure show decrements in the PV panel surface temperature when cooled by water and nanofluid, and the cooling effect is greater for the latter case. This result was indicated in all studies that used nanofluids in cooling PVT systems. However, the cooling efficiency differs; for example, for the case of the selected nanofluid, and for calculating the average temperature drop for a full day's operation, the temperature drop was around 57.5% and 17% compared to the two cases of the standalone PV module and water-cooled PVT system.

Figure 9 shows the change in efficiencies (electrical, thermal, and total) over time for the three systems tested. A standalone PV does not generate anything other than low electrical efficiency due to the high temperature of the PV panel. During the experiments, the current and voltage were measured from the beginning of the day (8 AM) to the evening (7 PM). The power values were found using Equation (3), including the determination of the electrical efficiency at each measurement period using Equation (5). The highest value of the electrical efficiency of this system reached 10% at 8:00 AM and started declining to reach the lowest value (4.5%) at midday. At 8:00 AM, the solar intensity was 119 W/m², which is insufficient to reach the maximum efficiency of the PV panel. When the solar intensity reached its maximum value of 1097 W/m² at 1:30 PM, the panel temperature reached 70 °C, which caused the efficiency deterioration mentioned. As for the water-cooled PVT (PVT$_w$) system, the electrical efficiency decreased at midday to reach its lowest value of 6.5%. When cooling the PVT system with the prepared nanofluid (PVT$_{nf}$), the lowest efficiency obtained was 9.07%. Compared to a full-day operation, the increase in the electrical efficiency of the nanofluid cooling system was 88.85% and 44% compared to the standalone PV and water-cooled PVT systems, respectively.

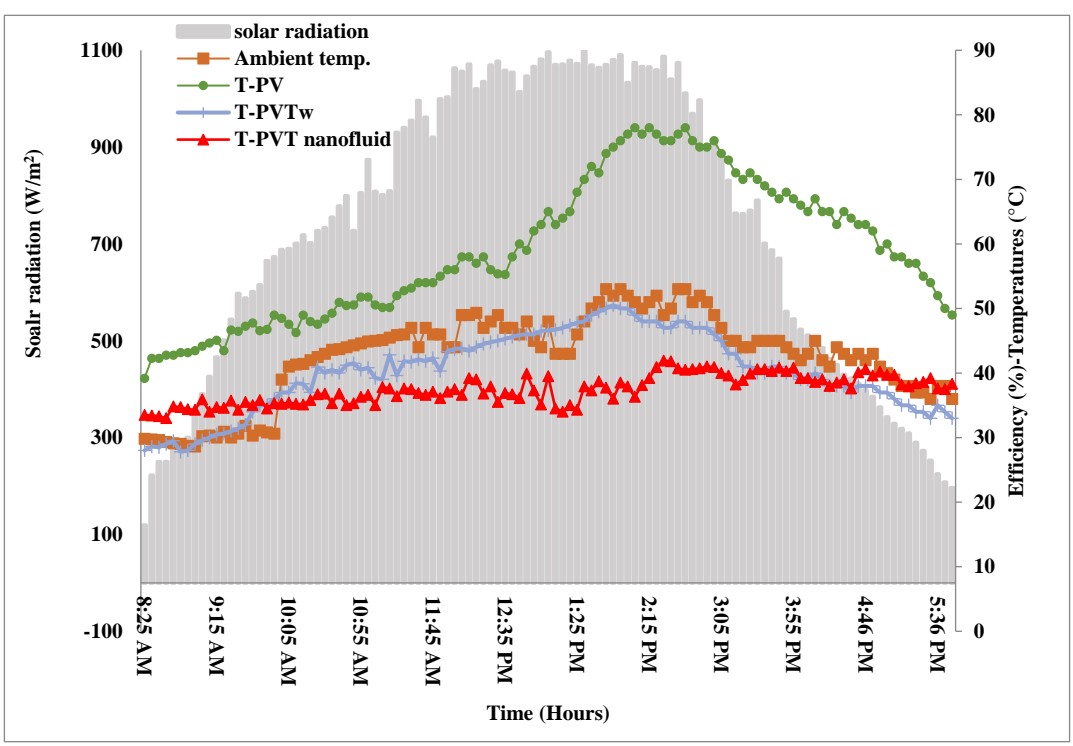

**Figure 8.** Solar radiation and temperature measured for the tested systems.

*4.5. PVT System Efficiencies*

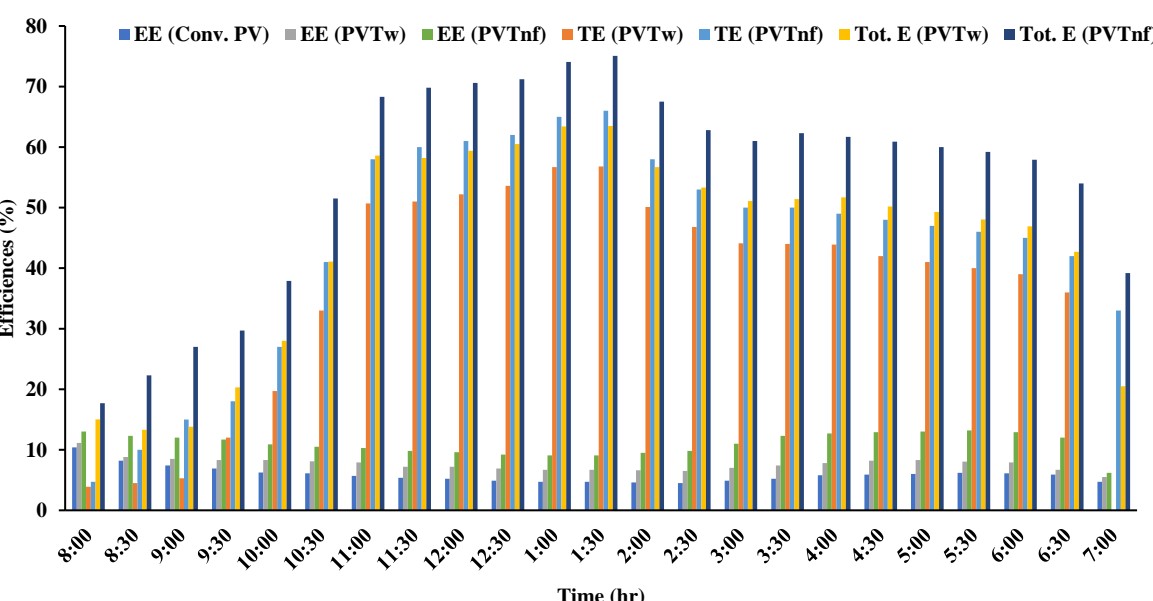

**Figure 9.** Efficiencies of the tested system variation with time.

Similarly, the useful accumulated heat was calculated using Equation (4) since the values of *To* and *Ti* were measured throughout the time and the water mass flow rate was predetermined. Using this temperature in Equation (6), the thermal efficiency can be obtained at each measurement period. As for the increase in thermal efficiency when cooling with the prepared nanofluid, it was higher than the water-cooling system by about 20%. While the electrical efficiency of all systems decreased at midday, the thermal efficiency of the nanofluid-cooled PVT system increased due to the high TC of this fluid.

The nanofluid-cooled PVT system achieved 75.08% maximum total efficiency, while the water-cooled PVT system's maximum total efficiency did not exceed 63.5%.

### 4.6. Comparison and Validation

In this part of the study, the results of the TC and ST of the prepared nanofluids are compared with other fluids from the literature. Additionally, a comparison of the performance of the proposed cooling nanofluid to the existing and conventional cooling methods for PVT applications is presented. Figure 10 compares the rate of TC enhancement. It must be emphasized here that such comparisons are not completely accurate, as there are clear differences in the type of nanoparticles added and thus in their TC and as a result of the prepared nanofluids' TC. Moreover, there are differences in the base fluid used, so this difference casts a shadow on the final TC of the nanofluid produced. In general, such figures indicate the appropriateness and scientific justification of the results. Comparing the results of the current study with the results of Ranjbarzadeh et al. in 2019 [19], Xing et al. in 2015 [40], Kazem et al. in 2021 [42], An et al. in 2016 [50], An et al. 2016 [60], khanjari et al. 2016 [61], Hjerrild et al. in 2016 [62], Khanjari et al. in 2016 [63], and Al-Ezzi et al. in 2022 [64] shows that the rate of improvement in the TC of the current study is higher compared to the rest of the studies. This can be considered a result of the TC of the MWCNT used and the optimum selection of the base fluid, surfactant, and sonication time. The optimum mixing produces a remarkable result and proves the correct study approach.

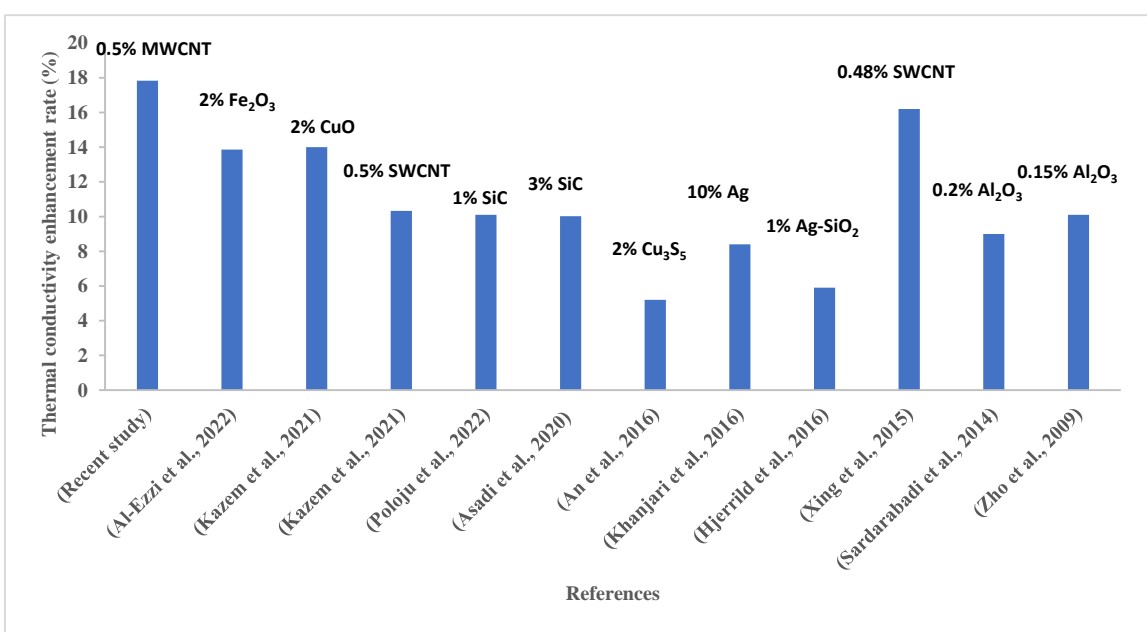

**Figure 10.** TC enhancement rate for the current study nanofluid compared to Poloju et al., 2022 [25], Xing et al., 2015 [40], Kazem et al., 2021 [42], Asadi and Alarifi, 2020 [46], An et al., 2016 [60], Khanjari et al., 2016 [61], Hjerrild et al., 2016 [62], Sardarabadi et al., 2014 [65], Kazem et al., 2021 [66], Al-Ezzi et al., 2022 [64], and Zho et al., 2009 [67].

Similarly, Figure 11 shows a comparison of the nanofluids prepared in this study ST with their counterparts from studies published in the literature by Chakraborty et al. in 2019 [68] and Sadeghi et al. in 2015 [69], a higher ST of the studied nanofluid is observed. The current studied nanofluid has a high ST, which proves the validity of the procedures taken during the tests and the effectiveness of the correct choices based on high-precision measurements. The results of the two figures show that the nano-emulsion consisting of water (base fluid), CTAB as a surfactant, and MWCNTs as additive nanoparticles are suitable to work in cooling PVT systems which require nanofluids with both high ST and TC.

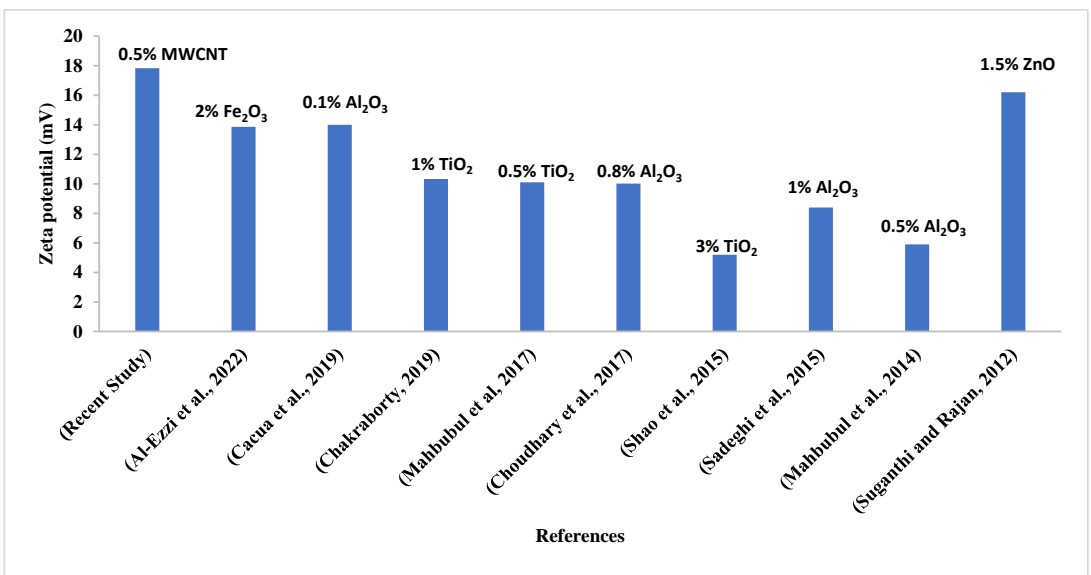

**Figure 11.** Zeta potential enhancement rate for the current study nanofluid compared to Mahbubul et al, 2017 [48], Chakraborty, 2019 [68], Sadeghi et al., 2015 [69], Al-Ezzi et al., 2022 [64], Cacua et al., 2019 [70], Choudhary et al., 2017 [71], Shao et al., 2015 [72], Mahbubul et al., 2014 [73], and Suganthi and Rajan, 2012 [74].

In Table 7, the electrical and thermal efficiencies of several systems from the literature are compared with the results of the current study. Certainly, this comparison will not be fair, because the systems used are for multiple PV panels with different efficiencies. The studies of [69] were also conducted in variable atmospheres than the current study. However, as mentioned previously, such comparisons give an indication of the validity of the approach taken in the current study and whether the results are similar to other studies or far from it. As for the electrical efficiency, the results show that what has been generated by the studied system is similar to what has been published in the literature. This result depends on the type of PV panel used, the solar radiation, and the panels' temperatures, and as previously explained, the environmental conditions of this study were the harshest in the world. As for the thermal efficiency, it is noted that the current PVT study system produced a high thermal efficiency that was only overcome by the systems of the studies of Venkatesh et al. in 2020 [41] and Aberoumand et al. in 2018 [75] who added expensive nano-silver was 4%, which means high costs, contrary to what the current study provided. As a final result, although the tests were carried out in very harsh weather conditions, what the PVT system produced in the current study was appropriate and efficient.

Figure 12 illustrates the comparison of PVT electrical and thermal efficiency in terms of PVT cooling methods. It is found that the thermal efficiency is relatively high and inconsistent with the literature results. Additionally, air cooling has the lowest efficiency ($\eta_e = 7.7\%$, $\eta_{th} = 28\%$) and nanofluid cooling has the highest efficiency ($\eta_e = 13.14\%$, $\eta_{th} = 68.22\%$). The air, water, and air/water cooling methods show the lowest efficiencies compared with nanofluid and/or nano-PCM cooling methods. All cost factors were considered when conducting the economic analysis, including civil and installation works, pump, heat exchanger, nanofluid, and mount. The costs for the system components were based on their local price, where the cost of the PV system was IQD 200 (priced at IQD 2/Wp): pump (IQD 40), heat exchanger (IQD 80), nanofluid (IQD 24, IQD 80/liter), pipe (IQD 20, IQD 1/m), and insulation (IQD 5). The proposed system costs about IQD 169 more than the conventional PV system. However, when analyzing the cost of energy, COE (which is the life cycle cost of the system divided by annual energy yield), we found that cooling using the proposed system results in a COE of IQD 0.027/kWh, while the conventional PV yields IQD 0.0338/kWh. This analysis was made using the simple life cycle cost and

COE model, also assuming only a 10% and 20% decrease in system yields in 10–20 and 20–25 years, respectively.

**Table 7.** Comparison of electrical and thermal efficiencies for the current PVT system study and others from the literature.

| Reference | Ref. No. | Added Nanoparticle | Base Fluid | Electrical Efficiency (%) | Thermal Efficiency (%) |
|---|---|---|---|---|---|
| Poloju et al. | [25] | 1.3% nano-$CuO_2$ | Water | 10 | 47 |
| Venkatesh et al. | [41] | 0.3% GNP | Water | 15 | 48 |
| Kazem et al. | [42] | 0.5% SWCNT | Water | 19 | 51 |
| Asadi and Alarifi | [46] | 0.3% MWCNT | Water | 9 | 45 |
| Li et al. | [76] | 0.48% Au | Water | 12.77 | 62.28 |
| Preet et al. | [77] | - | Water | 16 | 36 |
| Aberoumand et al. | [75] | 4% nano-Ag | Water | 11 | 70 |
| Sardarabadi et al. | [65] | 3% nano-$SiO_2$ | Water | 11 | 58 |
| Khanjani et al. | [63] | 5% nano-Ag | Water | 17 | 31 |
| Moradgholi et al. | [78] | 2% nano-$Al_2O_3$ | Methanol | 13 | 37 |
| Kazem et al. | [66] | 3% nano-SiC | 75% Water+ 25% EG | 20 | 43.3 |
| Al Ezzi et al. | [64] | 2% nano-$Fe_2O_3$ | 75% Water+ 25% EG | 12 | 60 |
| Menon et al. | [79] | 0.05% CuO | Water | 12.98 | 71.17 |
| Recent Study | - | 0.5 MWCNT | Water | 13.2 | 66 |

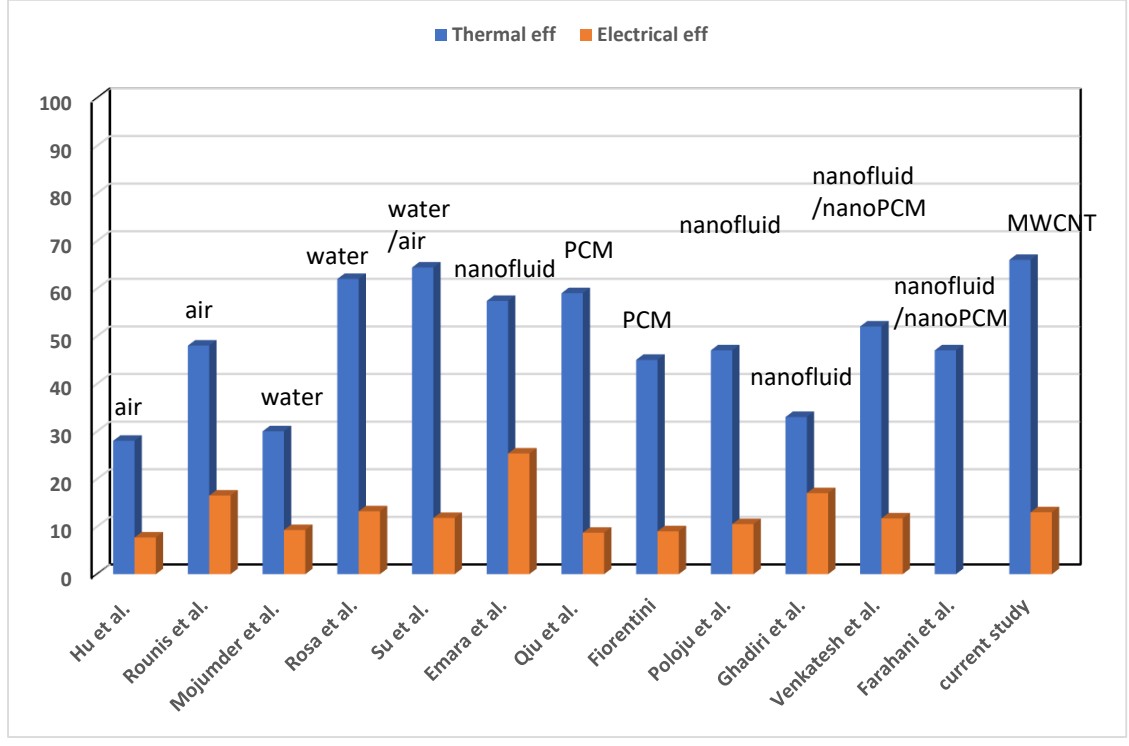

**Figure 12.** The performance of the proposed cooling nanofluid compared to Poloju et al., 2018 [25]; Venkatesh et al., 2022 [41], Hu et al., 2016 [80]; Rounis et al., 2017 [81], Mojumder et al., 2013 [82]; Rosa et al., 2016 [83]; Su et al., 2016 [84]; Emara et al., 2022 [85]; Qiu et al., 2015 [86]; Fiorentini et al., 2015 [87]; Ghadiri et al., 2015 [88], and Farahani et al., 2021 [89].

## 5. Discussion

In the previous work steps, four base fluids were used, and water was chosen (the best fluid found when MWCNT was added). However, this does not preclude the possibility of providing other base fluids with better thermophysical properties. Many references (such as Poloju et al. [25], Kazem et al. [77], and Al Ezzi et al. [64]) have indicated the preference for a mixture of water and ethylene glycol (75% to 25% volume fractions) in its distinguished characteristics, including that the nanoparticles, even after they are deposited at the rest of the nanofluid, return to mix with the base fluid when it is recirculated due to impact of ethylene glycol's lubricating property on the nanoparticles, preventing their adhesion to the inner surfaces of tubes. Therefore, it is preferable to conduct more experiments on several types of base fluids before recognizing that one specific base fluid is optimal for PVT applications. In this study, several surfactants were practically added to the base fluid to confirm the suspension of nanoparticles in the base fluid and to disrupt their sedimentation. In addition to this, suitable sonication timing was used for each addition ratio to confirm the same subject, and thus a nanofluid with high stability was reached. This stability should last for the long period of time in which the nanofluid remains stable, that is, without the deposition of a large part of the nanoparticles in the tubes, which causes material losses in addition to narrowing the paths and forming additional pressure on the pumps. The need to empty the nanofluid and re-mix it to maintain its high thermal conductivity and long-life stability will be at intervals when using the optimal preparation method. This gives the system stability and sustainability to provide a high performance during the required period.

The investigated PVT system, which produces electricity and heat, is used in many sustainable applications. PVT is used as a dryer for fruits, vegetables, animals, etc. (Çiftç et al. [90]; Kumar et al. [91]; Tiwari et al. [92]). Additionally, PVT is used for water desali nation (Kazem et al. [66]), biogas production (Su et al, 2021 [93]; Mglioli et al. [94]), building heating and ventilation applications (Migliol et al. [94]; Shao et al. [95]; Ramos et al. [96]), dehumidification (SolarVenti [97]), etc.

## 6. Conclusions

In this study, MWCNTs were selected as nanoparticles. These particles were mixed in varying mass fractions (0.1, 0.5, 0.75, and 1%) with four base fluids, which are water, ethylene glycol, propylene glycol, and heat transfer oil, to find the most suitable of these fluids to serve as a base fluid. Three types of surfactants were tested to find the best effect on the thermal conductivity and stability of the resulting fluid. Several sonication timings were also experimented to mix the base fluid, surfactant, and nanoparticles (1:30, 2:00, 2:30, 3:00, 3:15, 3:30, and 4:00 h). Experiments were carried out to reach the best nanofluid in terms of stability and thermal conductivity, taking into account the cost of this fluid. The results of this study reached the following conclusions:

1.  Water can be considered the best base fluid, as it has the highest thermal conductivity compared to the rest of the fluids. When 1% MWCNT mass fraction was added to the tested fluids, their TCs were increased. However, water + nano-MWCNT advanced all (TC, EG, PG, and oil-based nanofluids) by 119.5%, 308%, and 210%, respectively;

2.  Of the three types of surfactants studied (CTAB, SDS, and SDBS), it was found that CTAB gives the highest stability to the prepared nanofluids. For example, when 0.5% MWCNT was added to water, the rate of improvement in fluid stability was 10.3%, 5.2%, and 6.5% for adding SUR I, SUR II, and SUR III, respectively. When SUR I was added to 1% MWCNT + water, the improvement in ZP was 7.4% and 5.4% compared to using SUR II and SUR III, respectively;

3.  The effect of sonication time varies, as it is short when nanoparticles are added with a small mass fraction, and this time increases with an increase in the mass fraction. Adding 0.1% MWCNT to water required two and a half hours, while adding 0.5% MWCNT required a sonication time of three and quarter hours to achieve the maximum TC (0.67 W/m K). Additionally, the use of sonication time for a period

longer than the optimum one caused a decrease in the thermal conductivity and stability of the nanofluid.

A nanofluid was prepared from 0.5% MWCNT and, according to the best practices tested for preparation conditions, this fluid was circulated in a PVT system equipped to work in the external conditions of the city of Baghdad. The harshest weather conditions were chosen to test the effectiveness of the PVT system and the nanofluid. The results showed that despite the harsh external weather conditions, the system succeeded in maintaining a very appropriate electrical and thermal efficiency. The maximum electrical and thermal efficiencies achieved were 13.2% and 63%, respectively. By comparing the results of the current system with other studies, and despite the harsh conditions in which the tests were conducted, the results of this current study's PVT system were promising.

The results of this current study confirm the success of using the prepared nanofluid to cool the PVT system in the harsh weather of the city of Baghdad. However, there is still an urgent need to test many types of nanoparticles and prepare them in the same method used in this study and test them under the same harsh conditions. Achieving an optimal nanofluid dependence for use in PVT systems still needs further studies.

**Author Contributions:** Conceptualization, M.T.C. and A.A.A.-A.; methodology, M.K.S.A.-G.; validation, H.A.K., W.N.R.W.I. and M.S.T.; formal analysis, A.A.H.K.; A.J.A. and W.N.R.W.I.; investigation, K.S.; resources, K.S., W.N.R.W.I. and A.A.H.K.; data curation, M.K.S.A.-G., A.H.A.A.-W. and M.T.C.; writing—original draft preparation, M.T.C., A.J.A. and A.A.A.-A.; writing—review and editing, A.A.H.K., M.S.T. and A.H.A.A.-W.; supervision, M.T.C. and A.A.A.-A.; project administration, M.S.T., A.A.H.K. and M.T.C.; funding acquisition, H.A.K., W.N.R.W.I. and M.S.T. All authors have read and agreed to the published version of the manuscript.

**Funding:** This research received no external funding.

**Institutional Review Board Statement:** Not applicable.

**Informed Consent Statement:** Not applicable.

**Data Availability Statement:** Not applicable.

**Acknowledgments:** The authors acknowledge the Universiti Kebangsaan Malaysia (UKM) for their support under research code: GUP-2020-012.

**Conflicts of Interest:** The authors declare no conflict of interest.

### Nomenclature

| | |
|---|---|
| AC & $A_{module}$ | Collector and PV areas ($m^2$) |
| Cp | Water heat capacity (J/(K kg)) |
| G | Solar irradiance ($W/m^2$) |
| GS | Global solar radiation ($W/m^2$) |
| $I_{SC}$ & $I_{mp}$ | Short circuit and maximum point currents (A) |
| MF | Mass flow (kg/h) |
| PV | Photovoltaic |
| PVT | Photovoltaic/thermal |
| $P_{rated}$ & $P_{mp}$ | Rated and maximum point powers (W) |
| $T_{ambient}$ | Ambient temperature (°C) |
| TC | Cell temperature (°C) |
| $T_{in}$ and $T_{out}$ | Inlet and outlet temperature (°C) |
| $V_{OC}$ & $V_{mp}$ | Open circuit and maximum point voltages (V) |
| $W_R$ | Uncertainty |
| $\eta_{electrical}$ & $\eta_{thermal}$ | Electrical and thermal efficiencies (%) |

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
