# Peer review of "Effect of Different Preparation Parameters on the Stability and Thermal Conductivity of MWCNT-Based Nanofluid Used for Photovoltaic/Thermal Cooling"

_sustainability, doi:10.3390/su15097642_

Round 1

Reviewer 1 Report

1.                  The caption of Figure 1, 2 and Figure 3 should be revised for clear understanding.

2.                  Please mention the abbreviations of MWCNT in abstract.

3.                  This is a technical paper, authors should validate the work. However, please provide a comparison table with others relevant work.

4.                  Authors mentioned the electrical efficiency increase by 88.85% and 44%. How authors calculated the efficiency? Authors have mentioned the efficiency equations 3 to 7. However, where is the parameter values of those equations? Authors should mention all parameter values for a clear understanding of the efficiency percentage. Authors could mention the parameter value in the appendix section.  

Author Response

Dear reviewer,

We have revised our article according to your useful comments without any exception. However, below is our detailed respond to the received comments.

  1. The caption of Figure 1, 2 and Figure 3 should be revised for clear understanding.

Response: Done, the figures captions have been revised and modified for better understanding. Please refer to the revised manuscript.

  1. Please mention the abbreviations of MWCNT in abstract.

Response: Done, the abbreviation of MWCNT is add to the revised abstract. Please refer to the revised manuscript.

  1. This is a technical paper; authors should validate the work. However, please provide a comparison table with others relevant work.

Response: Done, Table 7 is added in validation section. Please refer to the revised manuscript.

  1. Authors mentioned the electrical efficiency increase by 88.85% and 44%. How authors calculated the efficiency? Authors have mentioned the efficiency equations 3 to 7. However, where is the parameter values of those equations? Authors should mention all parameter values for a clear understanding of the efficiency percentage. Authors could mention the parameter value in the appendix section.

Response: Done, all parameter values for a clear understanding of the efficiency percentage is added in the revised manuscript.

Thank you

Best regards

Reviewer 2 Report

Comments/Suggestions on the manuscript titled “Effect of different preparation parameters on the stability and thermal conductivity of MWCNTs-based nanofluid used for photovoltaic/thermal cooling.

The authors have investigated the effect of different preparation parameters on the stability and thermal conductivity of MWCNTs-based nanofluid used for photovoltaic/thermal cooling in details of Baghdad city. They reported that the nanofluid-cooled system achieved an electrical efficiency increase by 88.85% and 44% compared to standalone PV and water-cooled PVT systems, respectively. Also, its thermal efficiency was about 20% higher than that of a water-cooled PVT system. Overall, the authors have presented a nice piece of work, which would be of interest to the research/scientific community working on solar cell technologies and developments. Nevertheless, I do have following suggestions and minor comments:

Major comments:

1.     The authors should include the error bars wherever applicable.

2.      The authors should demonstrate reproducibility and long-term durability.

3.     The legend of the Figure is mistyped, i.e., “Ambient yemp.” should be “Ambient temp.”. Please correct it. Also, “T-PVT.w” should be “T-PVTw”. In addition, “T-PVTnanofluid” should be written with a space between PVT and nanofluid, i.e., “T-PVT nanofluid”

Minor comments:

4.     There are some grammatical errors in the manuscript, therefore the reviewer would suggest the authors to revisit all the sentences and rectify the errors carefully.

Author Response

Dear reviewer,

We have revised our article according to your respected comments without any exception. However, below is our detailed respond to the received comments.

Comments/Suggestions on the manuscript titled “Effect of different preparation parameters on the stability and thermal conductivity of MWCNTs-based nanofluid used for photovoltaic/thermal cooling.

The authors have investigated the effect of different preparation parameters on the stability and thermal conductivity of MWCNTs-based nanofluid used for photovoltaic/thermal cooling in details of Baghdad city. They reported that the nanofluid-cooled system achieved an electrical efficiency increase by 88.85% and 44% compared to standalone PV and water-cooled PVT systems, respectively. Also, its thermal efficiency was about 20% higher than that of a water-cooled PVT system. Overall, the authors have presented a nice piece of work, which would be of interest to the research/scientific community working on solar cell technologies and developments. Nevertheless, I do have following suggestions and minor comments:

Major comments:

  1. The authors should include the error bars wherever applicable.

Response: Done, error bars are added to figures 2 to 7 in the revised manuscript. Please refer to the revised manuscript.

  1. The authors should demonstrate reproducibility and long-term durability.

Response: Done, a statement on reproducibility and the long-term durability have been added to the manuscript in lines 310-320.

  1. The legend of the Figure is mistyped, i.e., “Ambient yemp.” should be “Ambient temp.”. Please correct it. Also, “T-PVT.w” should be “T-PVTw”. In addition, “T-PVTnanofluid” should be written with a space between PVT and nanofluid, i.e., “T-PVT nanofluid”

Response: Done, the figures’ legends are corrected in the revised manuscript.

Minor comments:

  1. There are some grammatical errors in the manuscript, therefore the reviewer would suggest the authors to revisit all the sentences and rectify the errors carefully.

Response: Done, the article language has been reviewed by professional proof-reading organization.

Thank you

Best regards

Reviewer 3 Report

Dear Authors

the paper is interesting and well structured, however I suggest some changes:

1. section 1 can be divided into two, creating section 2 . Literature review

2. Within section 1 should be a clear reference to the novelty of your work. So the reference to previous work already published and from these what gap emerged

3. The methodology section is perfect

4. In the results section I find the first values consistent with what the methodology proposes. Then proceed to present a section 3.6 whose title is not appropriate. I suggest creating a new section called Discussion. In this new section I suggest elaborating on some useful concepts to understand how your work moves toward sustainability.

5. Within the conclusion, do not be redundant in concepts and propose the limitations of your research.

6. I would improve within the paper the reference to the PV sector: two aspects that need to be framed within the literature. i) What drives the solar energy transition? ii) What is the critical parameter to improve PV panel performance?

Author Response

Dear reviewer,

We have revised our article according to your respected comments without any exception. However, below is our detailed respond to the received comments. 

the paper is interesting and well structured, however I suggest some changes:

  1. section 1 can be divided into two, creating section 2. Literature review

Response: Done, literature review section is added to the revised manuscript. Please refer to the revised manuscript.

  1. Within section 1 should be a clear reference to the novelty of your work. So the reference to previous work already published and from these what gap emerged

Response: Done. The discussion of the published studies in literature summarized to show the research gap. Please refer the lines 240-255. Also, the novelty of the current study is clarified in lines 256-267.

  1. The methodology section is perfect

Response: Thank you.

  1. In the results section I find the first values consistent with what the methodology proposes. Then proceed to present a section 3.6 whose title is not appropriate.

Response: Done, section 3.6 title is modified in the revised article.

  1. I suggest creating a new section called Discussion. In this new section I suggest elaborating on some useful concepts to understand how your work moves toward sustainability.

Response: Done, a new section titled Discussions is added to the revised manuscript, according to the reviewer's request.

  1. Within the conclusion, do not be redundant in concepts and propose the limitations of your research.

Response: Done, the conclusion is summarized and adhering to the limits of the research, according to the reviewer's request.

  1. I would improve within the paper the reference to the PV sector: two aspects that need to be framed within the literature. i) What drives the solar energy transition? ii) What is the critical parameter to improve PV panel performance?

Response: Done, the required aspects are added to the revised introduction. Please refer to the revised manuscript.

Thank you

Best wishes 

Round 2

Reviewer 3 Report

Dear authors the work still has some serious problems that have not been solved:

1. I do not notice improvements on the novelty of the work

2. check several errors present (for example there are 2 sections 2)

3. title section 3.6 still weak

4. discussion section is totally lacking in comparison with literature

5. I would improve within the paper the reference to the PV sector: two aspects that need to be framed within the literature. i) What drives the solar energy transition? ii) What is the critical parameter to improve PV panel performance? this comment is totally ignored.

If I search for What drives the solar energy transition? I see that the recent literature proposes. I don't see anything in your work

Author Response

Dear reviewer,

Thank you for your useful comments and suggestions,

Thank you

Best regards

Round 3

Reviewer 3 Report

All comments are well integrated. Congratulations.